# Isolation and Characterization of a Novel *Siphoviridae* Phage, vB_AbaS_TCUP2199, Infecting Multidrug-Resistant *Acinetobacter baumannii*

**DOI:** 10.3390/v14061240

**Published:** 2022-06-07

**Authors:** Meity Mardiana, Soon-Hian Teh, Ling-Chun Lin, Nien-Tsung Lin

**Affiliations:** 1Institute of Medical Sciences, Tzu Chi University, No. 701, Sec. 3, Zhongyang Rd., Hualien 97004, Taiwan; meitymardiana@gmail.com; 2Division of Infectious Diseases, Department of Internal Medicine, Hualien Tzu Chi Hospital, Buddhist Tzu Chi Medical Foundation, No. 707, Sec. 3, Zhongyang Rd., Hualien 97004, Taiwan; jimmyteh2000@gmail.com; 3Master Program in Biomedical Sciences, School of Medicine, Tzu Chi University, No. 701, Sec. 3, Zhongyang Rd., Hualien 97004, Taiwan

**Keywords:** *Acinetobacter baumannii*, bacteriophage, *Siphoviridae*, phage therapy

## Abstract

Multidrug-resistant *Acinetobacter baumannii* (MDRAB) is a pathogen recognized as antimicrobial-resistant bacteria involved in healthcare-associated infections. Resistance to antibiotics has made alternative therapies necessary. Bacteriophage therapy is considered a potential solution to treat MDRAB. In this study, we isolated and characterized the phage vB_AbaS_TCUP2199 (TCUP2199), which can infect MDRAB. Morphological analysis revealed that TCUP2199 belongs to the *Siphoviridae* family. TCUP2199 has a wide host range, can adsorb rapidly (68.28% in 2 min), and has a burst size of 196 PFU/cell. At least 16 distinct structural proteins were visualized by SDS polyacrylamide gel electrophoresis. A stability test showed that TCUP2199 was stable at 37 °C and pH 7. Genome analysis of TCUP2199 showed that it consists of a double-stranded DNA genome of 79,572 bp with a G+C content of 40.39%, which contains 98 putative open reading frames, none of which is closely related to the bacteriophage genome sequence that was found in the public database. TCUP2199 shows similarity in genomic organization and putative packaging mechanism with *Achromobacter* phage JWF and *Pseudoalteromonas* phage KB12-38 based on protein BLAST and phylogenetic analysis. Because of those unique characteristics, we consider TCUP2199 to be a novel phage that is suitable for inclusion in a phage cocktail to treat *A. baumannii* infection.

## 1. Introduction

*Acinetobacter baumannii* is a Gram-negative, aerobic, non-fermenting, catalase-positive, and oxidase-negative bacteria that is ubiquitous in the environment. *A. baumannii* has emerged as a salient pathogen due to its ability to cause healthcare-associated infections, representing one of the leading causes of nosocomial bloodstream infections in intensive care units. *A. baumannii* has been demonstrated to cause a number of clinical infections, including bacteremia, pneumonia, meningitis, and surgical site infections [1]. Initially, most antibiotics were effective against *A. baumannii*; however, the pathogen now appears to have become multidrug-resistant *A. baumannii* (MDRAB), which is defined as resistance to three or more antibiotics classes [2]. As a result of the large number of MDRAB reports coming from various countries, MDRAB has now become a global outbreak [3,4,5,6,7]. The high mortality of *A. baumannii* septicemia, which can reach up to 63.5%, has also become a severe problem that has to be addressed immediately [8,9,10,11].

Bacteriophages, also called phages, are the most abundant and diverse creatures on earth. A phage is a type of virus that infects bacteria. To kill bacteria, a phage injects its DNA or RNA into bacteria cells, eventually taking over bacteria machinery to replicate itself. Phage therapy has attracted a considerable amount of attention as a promising alternative to antibiotic treatment. Currently, phage therapy is used to effectively treat bacterial infection in clinical settings [12,13]. Phage cocktails have been administered intravenously and percutaneously into the abscess cavities of patients with diabetes, successfully rescuing the patient from a life-threatening MDRAB infection [14]. Moreover, the U.S. Food and Drug Administration recently approved the first clinical trial of intravenously administered phage therapy for ventricular assist device patients who have developed *Staphylococcus aureus* infection [15]. However, problems such as safety issues and narrow host range still limit the use of phage therapy. 

Phages are classified based on their morphology, and the largest order found in public databases is tailed phages (*Caudovirales*). They have three dominant families: phages with a non-contractile tail (*Siphoviridae*), phages with a contractile tail (*Myoviridae*), and phages with a short tail (*Podoviridae*), with proportions in the database of 66%, 20%, and 14%, respectively [16]. Although the *Siphoviridae* family represents the largest proportion of the phage in the database, only a few siphophages have been found to infect *A. baumannii* [17,18,19]. In this study, we isolated and characterized a novel *Siphoviridae* phage, TCUP2199, from wastewater around Hualien Buddhist Tzu Chi Hospital, Taiwan, which possesses a lytic ability to kill *A. baumannii*. We used sequence analysis to determine that this phage is completely different from previously identified phages and other siphophages [17,18,19,20]. To the best of our knowledge, this is the first study to reveal a wide host range of siphophages of *A. baumannii*.

## 2. Materials and Methods

### 2.1. Bacterial Strains and Growth Conditions

Strains of *A. baumannii* included in this study were as follows: 199 clinical isolates obtained from Taipei Veterans General Hospital (TVGH), 6 from Hualien Buddhist Tzu Chi Hospital (HBTZH), and 2 from ATCC. All bacteria were cultured on Luria–Bertani agar (LA) (Bacton, Dickinson and Company, Franklin Lakes, NJ, USA) at 37 °C for 18 h. One colony on LA was transferred to 5 mL of Luria–Bertani broth (LB) and then incubated at 37 °C with shaking at 150 rpm. *A. baumannii* growth was monitored turbidimetrically by measuring optical density at 600 nm (OD_600_), with an OD unit of 1.0 corresponding to around 3 × 10^8^ CFU/mL.

### 2.2. Antibiotic Susceptibility Test of A. baumannii

Random strains of *A. baumannii* (11 from TVGH, 5 from HBTZH, and 2 from ATCC) were selected to check their antibiotic susceptibility using an agar disk diffusion test against several antibiotics, including ciprofloxacin (5 µg), levofloxacin (5 µg), imipenem (10 µg), piperacillin-tazobactam (110 µg), and ampicillin (10 µg) (Becton-Dickinson, Sparks, MD, USA). The diameter of the inhibition zone was measured and classified according to the Clinical and Laboratory Standard Institute [21].

### 2.3. Phage Isolation

Wastewater samples collected from sewers around HBTZH were centrifuged (5000× *g* for 10 min at room temperature), and the supernatants were filtered through a 0.45 µm pore-size membrane to remove debris. Following overnight incubation, 28 random strains of *A. baumannii* clinical isolates from TVGH were employed to amplify the expected phage-containing supernatant and were then centrifuged and filtered through a 0.45 µm pore-size membrane. Strain TV2199 was found to strongly respond to filtered supernatant according to a spot test and was selected as an indicative host for phage propagation.

### 2.4. Purification of Phage Particles and Phage DNA

Crude phage lysates (200 mL, 10^9^ PFU/mL) were prepared by infecting TV2199 (as the host) with phages and incubating until the supernatant became clear. Crude lysates were centrifuged (8000× *g* for 10 min at room temperature), and the supernatant was passed through a 0.45 μm pore-size membrane filter and centrifuged at 18,000 rpm (Beckman Coulter Avanti JXN-26 centrifuge, JA-25.50 rotor; Beckman Coulter, Brea, CA, USA) for 2 h at 4 °C. The pellet was suspended in SM buffer (0.05 M Tris-HCl, pH 7.5, containing 0.1 M NaCl, 0.008 M MgSO4•7H2O, and 0.01% gelatin) and purified by banding on the block gradient of CsCl (ρ = 1.43, 1.45, and 1.5 g/cm^3^), followed by ultracentrifugation at 30,000 rpm (Beckman Coulter Optima XPN-100 ultracentrifuge, SW 41 Ti rotor; Beckman Coulter, Brea, CA, USA) for 3 h at 4 °C. The banded phage particles were recovered, dialyzed against double-distilled water, and stored at 4 °C until further use. Phage DNA was prepared according to methods described in [20], with some modifications. Briefly, phage suspension was treated with 1 µg/mL DNase I and 10 µg/mL RNase A (Promega, Madison, WI, USA) for 3 h at 37 °C. Then, the phage particles were concentrated with 20% PEG 6000 and 2.5 M NaCl, followed by centrifugation at 18,000 rpm (Beckman Coulter Avanti JXN-26 centrifuge, JA-25.50 rotor; Beckman Coulter, Brea, CA, USA) for 2 h at 4 °C. Finally, the concentrated phage encountered extraction with phenol/chloroform, and phage DNA was precipitated by ethanol.

### 2.5. Transmission Electron Microscopy

Phage morphology and attachment were examined by transmission electron microscopy (TEM) as previously described [20,22]. Briefly, a drop of 10^10^ PFU/mL of dialyzed phage particles was applied to the surface of a formvar-coated grid (300 mesh copper grids) (EMS, Hatfield, PA, USA), negatively stained with 2% uranyl acetate, and then examined with a Hitachi H-7500 TEM (Tokyo, Japan) operated at 80 kV. In order to observe the attachment of the phage to the host, the mixture of host cells with phage was incubated in 0.05 M phosphate buffer (pH 7.0) for 10 min and put on ice. Samples were taken, dropped on the grid, and stained with 2% uranyl acetate as above.

### 2.6. Host Range Analysis of phage

The host range of the phage was examined by spot test as previously described [12] on other 207 *A. baumannii* clinical isolates. Briefly, for each strain, overnight bacterial culture was mixed with melting 0.7% LB agar and overlaid onto an LB agar plate. Next, 5 µL of phage suspension (10^6^ PFU/mL) was spotted onto each of the bacteria lawns, which revealed clear zones for susceptible hosts after overnight incubation at 37 °C.

### 2.7. Phage Adsorption Curve

Bacterial cells were infected with phages to obtain an MOI of 0.0005 and incubated at 37 °C with shaking. Samples (100 µL) were taken at 0, 2, 4, 6, 8, 10, 15, 20, and 30 min and suspended in 0.9 mL of cold LB medium, followed by centrifugation at 12,000 rpm for 5 min (Heraeus Biofuge Pico microliter centrifuge, #3325 rotor; Heraeus instrument, Hanau, Germany) at room temperature. Samples were plated using the double-layer method to determine the unabsorbed phages titer. The phage adsorption efficiency was calculated with the following equation: (initial phage titer—unabsorbed phage titer in the supernatant)/initial phage titer multiplied by 100%.

### 2.8. Phage One-Step Growth Curve

Bacterial cells were infected with phages at an MOI of 0.01 and allowed to adsorb for 15 min on ice. The mixture was centrifuged (12,000 rpm, 10 min) with a Heraeus Biofuge Pico microliter centrifuge with a #3325 rotor, and the pellets containing infected cells were suspended in 20 mL of LB, followed by incubation at 37 °C. Samples were collected at 10-min intervals up to 80 min, immediately diluted in SM buffer, then poured using the double-layer method to determine the phage titer. The latent period and burst size were determined as described previously [23].

### 2.9. Lysis Curve and Optimal MOI Determination

To determine the bacteriolytic activity and optimal MOI of the phages, exponential growth-phase cultures of host cells were infected with phages at an MOI of 0.01, 0.1, 1, or 10 at 37 °C for 8 h in a 96-well microtiter plate (Thermo Fisher Scientific Inc., Waltham, MA, USA). The OD_600_ was measured every 30 min using a microtiter plate reader (Clariostar Plus, BMG Labtech, Offenburg, Germany). The experiments were performed in triplicate.

### 2.10. Pulse-Field Gel Electrophoresis (PFGE)

PFGE was performed for phage DNA size determination as described previously [24] using a CHEF-DR III system (Bio-Rad Laboratories, Hercules, CA, USA) at 5 V/cm with pulse ramps from 1 to 6 s for 20 h for undigested phage DNA at 14 °C in 0.5 × Tris-borate-EDTA buffer. A Midrange I PFG Marker (New England Biolabs, Ipswich, MA, USA) was used as the molecular size standard. 

### 2.11. Analysis of Phage Structural Proteins

Purified phage particles were mixed with sample buffer (1 M Tris-HCl pH 6.8 containing 10% sodium dodecyl sulfate, 20% glycerol, 1 M dithiothreitol, and 0.02% bromophenol blue) and heated in boiling water for 10 min, followed by separation of the protein in SDS polyacrylamide gel (10%) electrophoresis.

### 2.12. Thermal and pH Stability Test

The stability of phages under certain thermal and pH conditions was determined according to the protocol described in [25], with some modifications. Briefly, the phage suspension was incubated at 37, 50, and 70 °C for 24 h, and samples were collected at 0, 1, 5, 10, and 24 h. For determination of pH stability, the phages were incubated at pH 3, 5, 7, 9, and 11 for 24 h at 37 °C. The phage titer from collected samples was determined by a double-layer method. All experiments were performed in triplicate. 

### 2.13. DNA Sequencing and Genome Analysis

Phage DNA genome was sequenced by Genomics BioSci & Tech (Taipei, Taiwan). Potential open reading frames (ORFs) of the phage genome were scanned with ORF Finder (https://www.ncbi.nlm.nih.gov/orffinder/; accessed on 12 December 2020) and GeneMarks software [26]. The genome sequence was aligned and compared with that of other phages using BLASTn. The gene function of translated ORFs was predicted by BLASTp [27] and HHpred [28]. The presence of tRNAs in the genome was examined by tRNAscan-SE [29]. TMHMM Server, v. 2.0, was used to predict transmembrane protein. Phylogenetic analysis was performed using the neighbor-joining method with 1000 bootstrap replication in MEGA X [30]. A comparative genomic study was conducted by VIPTree [31] and Easyfig [32]. The genome sequence of TCUP2199 was deposited in GenBank under the accession number ON323491.

## 3. Results

### 3.1. Antibiotic Susceptibilities of Clinical Isolates of A. baumannii

A pathogen is classified as an MDRAB if it is non-susceptible to at least one agent in three or more antimicrobial categories [33]. In the present study, we selected 11 clinical isolates from TVGH, 5 from HBTZH, and 2 from ATCC and checked their susceptibility profile against several antibiotics. As shown in Table 1, 8 out of 18 strains were considered MDRAB. Some strains were also found to have intermediate resistance to some antibiotics with increased dosage [34].

### 3.2. Isolation, Purification, Morphology, and Host Range Analysis of TCUP2199

Wastewater samples from sewers around Buddhist Tzu Chi Hospital were enriched and screened by spot test on the lawn of *A. baumannii* clinical isolates. One phage responding to TV2199 was isolated and named TCUP2199. This phage formed very small plaques (average size: 0.16 ± 0.2 mm), and no surrounding translucent halo was observed after incubation at 37 °C for 24 h on the bacterial lawn (Figure 1a). In addition, TEM analysis was conducted to check the morphology of phage TCUP2199, as well as phage–host attachment. The results showed that TCUP2199 has an icosahedral head (~78.6 nm) and a long, non-contractile tail (~380 nm) (Figure 1b). Figure 1c shows that TCUP2199 attached to the bacterial surface through its tail, and the tail could also face away from the bacterial surface sometimes. Based on the morphology, TCUP2199 was determined to belong to order *Caudovirales* and family *Siphoviridae*.

During phage particle purification, TCUP2199 lysate (10^11^ PFU) was subjected to ultracentrifugation in a discontinuous CsCl gradient (1.43, 1.45, and 1.5 g/cm^3^) and was found to band in 1.45 g/cm^3^ blocks, suggesting that TCUP2199 has a buoyant density of between 1.45 and 1.50 g/cm^3^ (Figure 2a). We estimated a TCUP2199 genome size of approximately 82 Kb by PFGE (Figure 2b). Analysis of the structural proteins of the phage particle through SDS-PAGE showed four major bands and more than 10 minor protein bands, with molecular weights ranging from 20 to 100 kDa (Figure 2c).

The host range of TCUP2199 was determined using 208 clinical isolates of *A. baumannii*, including TV2199. The result showed that TCUP2199 lysed 90.86% of the tested isolates (Appendix A). TCUP2199 can infect more clinical isolates than our previously collected phages [20], meaning that TCUP2199 is suitable for controlling *A. baumannii* infection. 

### 3.3. Replication Kinetics and Lytic Activity of TCUP2199

The absorption rate of phage TCUP2199 onto TV2199 was investigated, and the results showed that approximately 68.28% of the phage particles were adsorbed within 2 min, with more than 98% being absorbed after 20 min (Figure 3a). The infection dynamics of TCUP2199 were determined by a one-step growth curve. The latent period was about 30 min. The burst size of the phage, calculated by the ratio of the liberated phage to the originally infected bacteria during the latent period, was approximately 196 PFU/cell after the latent period (Figure 3b).

To evaluate the bacteriolytic effect and optimal MOI of TCUP2199, TV2199 was either not infected or infected with TCUP2199 at different MOIs, and bacterial growth was observed by optical density monitoring. In the present study, optimal MOI was defined as the lowest concentration at which a phage can effectively inhibit host cell growth. The results showed that the turbidity of uninfected TV2199 increased continuously, and the optimal MOI was determined to be 1.0, as this concentration can sufficiently maintain the growth of cells to the initial point (Figure 3c).

### 3.4. Thermal and pH Stabilities of TCUP2199

To assess the stability of TCUP2199, the phage was exposed to various temperatures. The result showed that TCUP2199 was stable at 37 °C. When incubated at 50 °C, TCUP2199 caused the phage titer to decrease to almost 50% after 10 h, and no activity was shown when the TCUP2199 was exposed to a temperature of 70 °C within 1 h (Figure 4a). TCUP2199 encountered no infective ability at pH 3, with decreased activity at pH 5 and 11 and remaining stable at pH 7 (Figure 4b).

### 3.5. Whole-Genome Analysis of TCUP2199

Sequencing of TCUP2199 revealed that the genome consisted of 79,572 bp with 127-bp repeats at both ends, which is, overall, similar to the size shown by PFGE (Figure 3b), with a GC content of 40.39%. No tRNAs were detected in the genome. BLASTn analysis revealed that no *A. baumannii* phage shares similarity with the TCUP2199 whole genome. Most of the genes (75.5%) were located on the same strand, and they were aligned neatly from the end of the genome, whereas only 24 genes (24.4%) were located on the opposite strand (Figure 5). Only about 27.5% (27 out of 98) of the predicted ORFs are similar to genes with known function in the GenBank database, whereas the remaining 71 ORFs were hypothetical proteins or not matched to any functional proteins. Based on the conserved domain analysis, the annotated proteins of TCUP2199 were categorized into several groups: those associated with DNA replication, recombination, repair, processing and metabolism, lysis system, DNA packaging, and structural proteins (Figure 5, Appendix A). 

Genes identified to encode DNA replication, recombination, repair, processing, and metabolism include DNA-binding HTH domain protein (ORF41), DEAD-like helicase (ORF41), DNA polymerase I (ORF44 and ORF56), exonuclease (ORF61), metallo-dependent phosphatase (ORF64), nucleoside hydrolase (ORF65), thymidylate synthase (ORF71), ribonucleoside-diphosphate reductase (ORF73), ribonucleotide reductase (ORF74), DNA-binding protein (ORF93), and primase (ORF97). Structural proteins were identified, such as capsid proteins (ORF 46 and 91), tail fiber proteins (ORF47 and 76), tail protein (ORF75), tail assembly proteins (ORF77 and 78), minor tail protein (ORF80), and tail tape measure protein (ORF80), which were distributed throughout the whole TCUP2199 genome. Terminase large subunit protein was also identified on the genome and was classified as DNA packaging protein. All known protein functions were identified by BLASTp, with the identity of their amino acid sequences ranging from 26 to 63%. Global comparisons to other phages were carried out using VIPTree and revealed that no closely related genome matched the TCUP2199 genome, with distant similarities to at least three other phages, such as *Pseudomoalteromonas* phage KB12-38, *Achromobacter* phage JWF, and *Caulobacter* phage Seuss (Figure 6a). TBLASTX analysis showed several ORFs of TCUP2199 similar to the ORFs of the other two phages and their genomic organization at the right-arm part of TCUP2199 (encompassing from ORF41 to ORF97) (Figure 5 and Figure 6b). These ORFs include functional modules involved in DNA processing, tail morphogenesis, and packaging systems (Appendix A).

### 3.6. Phylogenetic Analysis of TCUP2199

Phylogenetic analysis was carried out with terminase large subunit protein (encoded by ORF95) as sample protein compared with other phages using neighbor-joining analysis and MEGA X software to construct the tree. Terminase large subunit is considered the most universally conserved gene in phages; thus, it is frequently used for phage classification in phylogenetic analysis [35]. Based on the limited datasets, *Enterococcus* phage EfV12-phi1 was selected as an outgroup because it belongs to *Myoviridae*. As shown in Figure 7, TCUP2199 was clustered with *Pseudomoalteromonas* phage KB12-38, *Achromobacter* phage JWF, and *Caulobacter* phage Seuss, which belongs to the *Siphoviridae* family, as also confirmed by the transmission electron micrograph shown above.

## 4. Discussion

The resistance of *A. baumannii* to multiple antibiotics limits the patient’s ability to recover and prolongs their stay in hospital. Therefore, the discovery and development of alternative therapies to treat MDRAB is highly warranted. In this study, we successfully isolated TCUP2199, which belongs to *Siphoviridae* and has a long and non-contractile tail. TCUP2199 has the longest tail compared to other previously found *A**. baumannii* siphophages [17,18,19]. Figure 1c shows that TCUP2199 embedded its tail in the bacterial surface to establish initial interaction. Conventionally, bacteriophages employ their tail to recognize and attach to the host cell, after which they inject their DNA from the capsid into the cytoplasm of the host cell [36,37]. TCUP2199 also used its head to approach the host surface. Although it seems unusual, this phenomenon has been reported in Wip1 bacteriophage, which infected *Bacillus anthracis*. This might be due to the labile vertex undergoing tubular transformation or reversible binding between the phage tube-like structure and its host [38].

The first step in phage infection is recognition and binding to the host, which is mediated by phage receptor-binding proteins (RBPs). RBPs have been identified as tail fiber proteins that determine the host specificity. Two similar phages, *P. aeruginosa* phages PaP1 and JG004, share high DNA sequence homology but exhibit host specificities. A single-point mutation in the putative tail fiber gene of JG004 yields a broader host range than the parental phage, and the replacement of the tail fiber gene of PaP1 with the corresponding gene in JG004 also altered its host range [39]. A similar result was observed in chimeric *A. baumannii* phages ϕAB1, in which the tail fiber gene was replaced by the tail fiber gene of ϕAB6, resulting in a change of host specificity [40]. Multiple tail fiber proteins allow phages to infect and replicate in various bacterial strains [41]. TCUP2199 has approximately three tail fiber proteins (ORF47, ORF75, and ORF76), which we suggest increase its host range compared to other *A. baumannii* phages previously discovered in our laboratory [20]. More detailed studies are needed to fully understand whether each ORF recognizes different bacterial capsule types and infects more strains of *A. baumannii*.

Our comparative genome study showed two phages (*Achromobacter* JWF and *Pseudoalteromonas* KB12-38) have similar organization and share some ORFs with TCU2199 (Figure 6). In the *Achromobacter* JWF phage [42], based on the sequence of the terminase large subunit, JWF did not form a single branch in a phylogenetic tree and showed protein similarity with *Bacillus* phage SPO1; therefore, we conclude that JWF might have a similar packaging mechanism as that of *Bacillus* phage SPO1. We also performed a similar phylogenetic analysis based on terminase sequences from different packaging mechanisms. The result revealed that TCUP2199 ORF95 clustered together with *Achromobacter* JWF and *Pseudoalteromonas* KB12-38, with a significant bootstrap value (Appendix A). This suggests that these three phages might share the same packaging mechanism.

Phages are useful as an alternative therapy for bacterial infection because they encode several enzymes that can treat bacterial infection and biofilms, such as depolymerase and hydrolases [43]. Endolysin is a peptidoglycan hydrolase employed by the phage to degrade the peptidoglycan layer of the host so that at the end of lytic cycle, the phage can burst out of the bacteria [44]. ORF52 has 50% similarity with the peptidoglycan-binding protein of *Acinetobacter phage*, vB_AbaM_ME; therefore, ORF52 was identified as a putative endolysin. Along with endolysin, holin also plays a pivotal role in the cell lysis system, as it creates holes and releases endolysin to hydrolyzed cell walls to release phage progeny out of the host [33,45,46]. We previously predicted ORF50 as a putative holin as it is a small protein with four transmembrane α-helical segments located upstream of endolysin [45,47]. Further study is warranted to confirm the ability of TCUP2199-derived endolysin protein to treat *A. baumannii* infection or to disperse its biofilm.

The development of phage resistance by *A. baumannii* has been reported in many studies [48,49]. In this study, this phenomenon also occurred after long-term culturing (Appendix A). This might be due to selection pressure by the phage, resulting in the retention of some cells with bacteriophage-resistant phenotypes during exponential growth and prevention of their disruption by the phage [49]. The other reason might that TCUP2199 follows the lysogenic life-cycle-like phage JWF, which also features a broad host spectrum [50]. Although it seemed to weaken the application of TCUP2199, this phage could still be a good choice as a first-line treatment for bacterial infection, considering its wide host range. In a previous study, phage-resistant bacteria exhibited increased sensitivity to antibiotics [51]. This exciting premise might also be proven in the future.

In this present study, we compared the host range of TCUP2199 with that of other phages previously discovered in our laboratory [20] and found that TCUP2199 has the broadest spectrum of all. This characteristic makes TCUP2199 valuable for treating clinical isolates of *A. baumannii* infection. We tested its antibiotic susceptibility on 18 random clinical isolates of *A. baumannii* in this study, and most of them were determined to be MDARB that can also be infected by TCUP2199. Thus, we can conclude that TCUP2199 is a good candidate for phage therapy to treat MDRAB. 

The present study also revealed unique characteristics of TCUP2199, such as large burst size and low similarities with related phages in the database; therefore, it can be considered a novel phage. The phenomenon of the development of phage resistance or lysogeny suggests that the application of TCUP2199, along with other lytic phages, in cocktails is needed to eradicate *A. baumannii* infection.

## Figures and Tables

**Figure 1 viruses-14-01240-f001:**
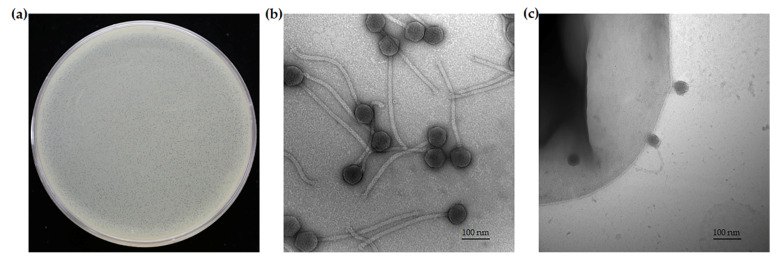
Morphology of TCUP2199 listed as (**a**) plaques of TCUP2199 on TV2199 bacterial lawn; (**b**) electron micrograph of *A. baumannii* TCUP2199 with 200,000× magnification; (**c**) attachment of TCUP2199 to TV2199 with 150,000× magnification. TCUP2199 and TV2199 were negatively stained with 2% uranyl acetate. The bar corresponds to 100 nm.

**Figure 2 viruses-14-01240-f002:**
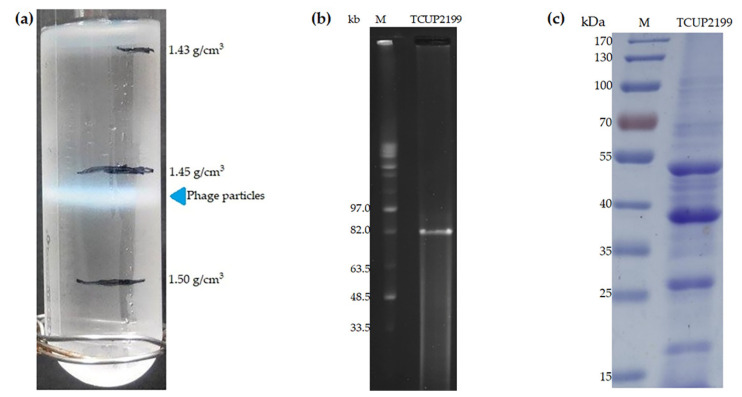
Purification, DNA extraction, and structural proteins of TCUP2199. (**a**) Block gradient of CsCl. The band indicates phage TCUP2199 particles. (**b**) Estimation of TCUP2199 DNA size by pulsed-field gel electrophoresis. Lanes: M, midrange I PFG markers; second lane indicates uncut TCUP2199 DNA. (**c**) SDS polyacrylamide gel electrophoresis (10%) of TCUP2199 virion proteins. Purified TCUP2199 particles were boiled in sample buffer (100 mM Tris-HCl, pH 6.8, 4% SDS, 0.2% bromophenol blue, 20% glycerol, 200 mM dithiothreitol) (10 µL) and loaded onto the well.

**Figure 3 viruses-14-01240-f003:**
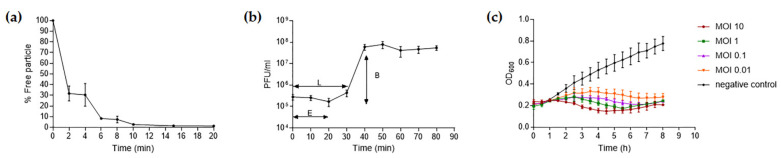
Biological properties of TCUP2199. (**a**) Adsorption of TCUP2199 to TV2199. Approximately 70% of TCUP2199 particles were adsorbed onto the cells after 2 min, and nearly all phage particles were absorbed within 20 min. (**b**) One-step growth of TCUP2199 on TV2199. The eclipse and latent period of TCUP2199 were approximately 10 min and 30 min, respectively, and the burst size was 196 PFU/cell. L: latent period; E: eclipse period; B: burst size. (**c**) Lytic activity of phage TCUP2199 against TV2199 at different MOIs. All experiments were performed in triplicate; error bars represent the standard error of the mean (SEM).

**Figure 4 viruses-14-01240-f004:**
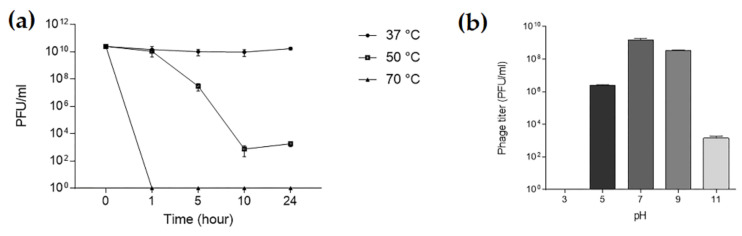
Thermal and pH stability test of TCUP2199. (**a**) TCUP2199 stability at various temperatures; samples were taken at 0, 1, 5, 10, and 24 h. (**b**) TCUP2199 stability at various pH values; samples were kept at 37 °C for 24 h. The sample titers were checked using a double-layer method. The experiments were performed in triplicate, and the data are shown as the mean ± SEM.

**Figure 5 viruses-14-01240-f005:**
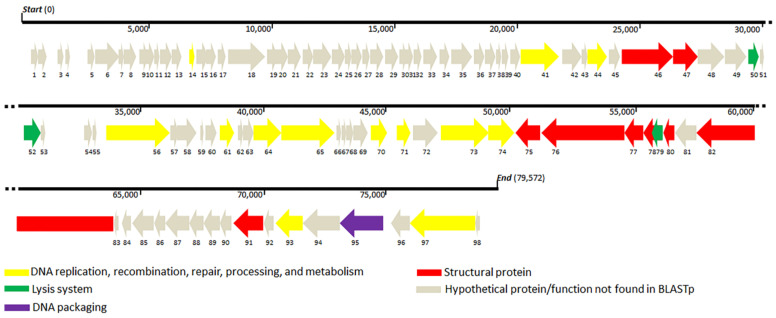
Representation of open reading frame (ORF) (ORF1 to 98) organization of TCUP2199. The predicted genes are indicated by arrows of different colors.

**Figure 6 viruses-14-01240-f006:**
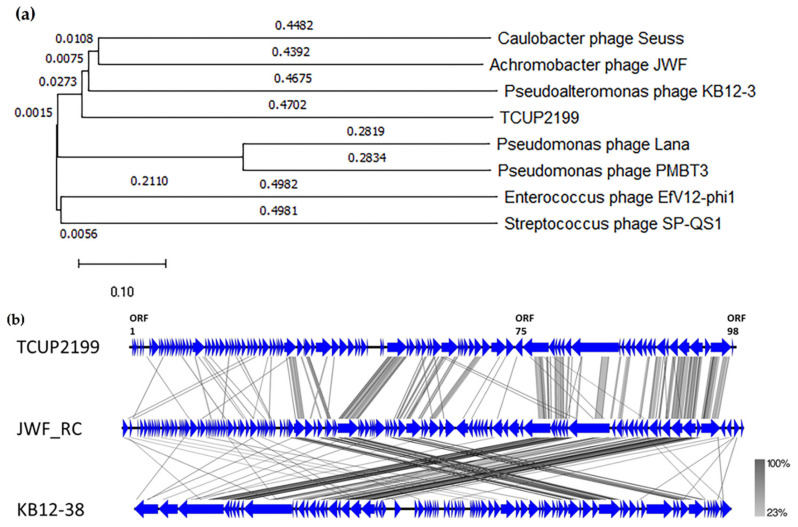
Genome comparison of TCUP2199 and related phages. (**a**) Genome similarity tree of TCUP2199 to other phages by VIPTree. (**b**) Comparative genomic organization of phage TCUP2199 and related phages. JWF_RC represents the reverse-complemented genome organization of *Achromobacter* phage JWF (NC029075); KB12-28 represents the genome of *Pseudomoalteromonas* phage KB12-38 (MF098558). The figure was generated using Easyfig by TBLASTX. The gradient scale indicates the similarity range.

**Figure 7 viruses-14-01240-f007:**
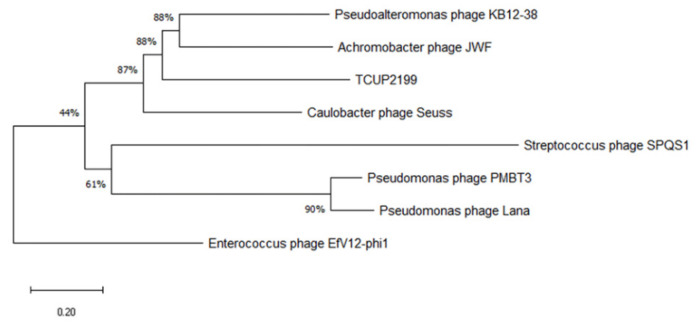
Phylogenetic tree analysis of relatedness of TCUP2199 among other phages based on their terminase large subunit protein. The amino acid sequences were aligned using MUSCLE, and the tree was constructed using neighbor-joining analysis with 1000 bootstrap replications.

**Table 1 viruses-14-01240-t001:** Agar diffusion test of TV2199.

Strain	Antibiotic Disc-Zone Diameter (mm)				
	CIP (5 µg)	LVX (5 µg)	IPM (10 µg)	TZP (110 µg)	AMP (10 µg)
TV479	24	11	26	17	0
TV481	25	27	28	22	9
TV530	24	25	30	22	0
TV574	26	26	30	23	0
TV962	24	25	27.5	17	0
TV967	24	23	28	20	9
TV1733	0	11	23	17	0
TV2050	12	11	20	17	0
TV2177	25	25	13	20	0
TV2199	9	14	25	13	0
TV2606	24	24	28	20	0
8 -- 2	0	12	13	9	0
9 -- 3	0	11	14	16.5	0
M3237	0	0	13	14	19
M6777	0	12	0	9	0
M68316	0	7	14	14.5	0
ATCC17978	26	23	27	23	0
ATCC19606	26	26	32	26	0

Color legend: red: resistant; blue: sensitive; green: intermediate.

## Data Availability

The data presented in this study are available in this article and Appendix A.

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
