# Peer review of "Isolation and Characterization of a Novel Siphoviridae Phage, vB_AbaS_TCUP2199, Infecting Multidrug-Resistant Acinetobacter baumannii"

_viruses, 2022, doi:10.3390/v14061240_

Round 1

Reviewer 1 Report

L68, “wide host range”, need more data support this. Table S1 only showed host range of clinical isolates from one hospital which may share similar genetic background, how about other standard strains?

L80, only one isolate was picked? How about the antibiotic susceptibilities of other strains?

L93, does TV2199 happen to be the chosen one in L80? I suppose the TV2199 was picked after phage isolation, pleased describe more clearly in the manuscript.

L130, why only MOI of 0.005 was used?

L183-184, this sentence is suggested to be merged into 2.13.

L186, “Antibiotic susceptibilities of A. baumannii TV2199” would be better.

L234, it is suggested to test the optimal MOI in biological characterization of TCUP2199.

Figure 3(c) only showed data in 8h, did author test the OD600 after 12-48h? I suggest authors perform the lytic experiment from 0-48h.The OD600 may elevate after 24h, which can weaken the application prospect of TCUP2199.

L256, “Thermal and pH Stabilities of TCUP2199” would be better.

Global comparison (for example the VIPTree) of TCUP2199 genome to other phages should be carried out.

L412&418, Acknowledgments repeated twice.

Though the manuscript is well written and organized, I still recommend the authors seek help from language professionals to correct the grammar errors and the choice of words/terms in the manuscript. 

Reviewer 2 Report

This is a very interesting paper describing the identification and characterization of a novel phage against A. baumannii infection. Regarding the urgent need for such phages that could be utilized against antibiotic resistant infections, I recommend publication of this manuscript. Before publication, however, the manuscript should be carefully checked for language (there are some grammatical and other mistakes). One additional point: Line 47: “injects (…) into its capsid“ – do you mean that the genome is injected into the bacterium?
